# The Effect of Various Forms of Sulfur on Soil Organic Matter Fractions and Microorganisms in a Pot Experiment with Perennial Ryegrass (*Lolium perenne* L.)

**DOI:** 10.3390/plants12142649

**Published:** 2023-07-15

**Authors:** Elżbieta G. Magnucka, Grzegorz Kulczycki, Małgorzata P. Oksińska, Jolanta Kucińska, Katarzyna Pawęska, Łukasz Milo, Stanisław J. Pietr

**Affiliations:** 1Laboratory of Biogeochemistry and Environmental Microbiology, Department of Plant Protection, Wrocław University of Environmental & Life Sciences, Grunwaldzka St. 53, 50-357 Wrocław, Poland; malgorzata.oksinska@upwr.edu.pl (M.P.O.); jolanta.kucinska@upwr.edu.pl (J.K.); stanislaw.pietr@upwr.edu.pl (S.J.P.); 2Institute of Soil Science, Plant Nutrition and Environmental Protection, Wrocław University of Environmental & Life Sciences, Grunwaldzka St. 53, 50-357 Wrocław, Poland; grzegorz.kulczycki@upwr.edu.pl; 3Institute of Environmental Engineering, Wrocław University of Environmental & Life Sciences, Grunwaldzki Sq. 24, 50-363 Wrocław, Poland; katarzyna.paweska@upwr.edu.pl; 4Chemical Plants “Siarkopol” Tarnobrzeg Ltd., Chemiczna St. 3, 39-400 Tarnobrzeg, Poland; lmilo@zchsiarkopol.pl

**Keywords:** sulfur and sulfate fertilizers, ryegrass, soil organic matter, water-soluble organic carbon, humic acids, glomalin-related soil proteins

## Abstract

This article focuses on the agronomic evaluation of the supplementation of mineral NPKMg fertilizers with elemental sulfur, magnesium, potassium, or ammonium sulfates in pot experiments with ryegrass growing in a sandy Arenosol with very low sulfur content. A benefit evaluation was carried out on the basis of biomass production, crop nutritional status, and changes in the content of soil organic matter fractions. Furthermore, the total number of bacteria, nitrogen-fixing bacteria, and fungi was estimated using the qPCR technique in soil samples after 60 days of ryegrass growth. The combined application of NPKMg and sulfur or sulfate fertilizers significantly increased the summary yield of three cuttings of fresh ryegrass biomass in the range of 32.3% to 82.7%. The application, especially in the form of sulfates, significantly decreased the content of free phenolic acids. Furthermore, compared to the control, i.e., soil with NPKMg applied alone, an increase in the content of glomalin-related proteins and a decrease in the amount of water-soluble organic carbon compounds were observed. Neither the number of DNA marker copy numbers of the total bacterial community nor the nitrogen-fixing bacteria were noticeably different. In turn, the total number of genetic markers for fungi was significantly higher in soils with potassium or ammonium sulfates compared to the control soil. The general results suggest that the application of sulfur fertilizers with NPKMg mineral fertilizer can benefit crops and support soil fertility due to the stabilization of aggregates and the decrease in water-soluble organic compounds.

## 1. Introduction

In natural ecosystems and agroecosystems, sulfur is one of the most important macronutrients that ensure the normal growth and development of many plants and microorganisms. Sulfur is mainly a component of cysteine and methionine, making it critical for the biosynthesis of specific proteins, the essential components of all living organisms. In plants, it is a constituent of glutathione, phytochelatins, coenzyme A, and vitamins such as biotin and thiamine. Sulfur-containing compounds are primarily required in photosynthesis, respiration, and energy generation and are crucial for plant survival during biotic and abiotic stress, e.g., [1,2,3]. In addition, a sufficient amount of sulfur in the soil is required for the proper maintenance of the soil organic matter (SOM) stocks and soil fertility [4,5,6]. In the last three decades, the above 85% reduction in industrial sulfur emissions into the atmosphere was mainly in North America and Europe [7,8], resulting in its limited availability in several regions. Since the 1990s, sulfur has become one of the most limiting nutrients for agricultural production in many European countries [9,10]. The deficiency of this element has occurred on approximately 100 × 10^6^ ha of arable land in different parts of the world, and the use of sulfur fertilizers containing sulfate or elemental sulfur is now essential for successful crop production [11]. Sulfate is the main source of this nutrient for plants; in turn, elemental sulfur requires microbial oxidation to this anionic form for its absorption by the root system [4,5,6]. Importantly, the shortage of mineral sulfur in the soil results in the microbial degradation of SOM. This fraction serves as this nutrient reservoir for plants and microbes because, in general, more than 95% of soil S is organically bound [12,13,14]. Organic sulfur compounds in the soil are mainly sulfate ester and carbon-bonded S, and their conversion to mineral forms involves biochemical and biological mineralization, respectively; see, e.g., [15,16,17]. The hydrolysis of the ester-bonded S fraction of SOM by extracellular sulfatase enzymes in plant roots or microbes increases if a low amount of mineral S-SO_4_ occurs in the soil solution, while the oxidation of the C-bonded S fraction requires direct microbial activity and is driven by its need for organic carbon to provide energy [17,18]. According to this model, factors such as the plant species [6] and S fertilizer can affect the degradation process of SOM [19,20,21]. Thus, the appropriate application of sulfur fertilizers, such as elemental sulfur and sulfate forms, mainly as salts of ammonium, magnesium, or potassium, is an important treatment that improves crop productivity [11,22] because the availability of sulfur determines the efficacy of nitrogen fertilizers and consequently affects plant composition and quality, e.g., [23,24,25,26,27]. The described positive effect of sulfur fertilization is related to increases in nitrogen uptake, and plants’ protein nitrogen is the dominant fraction, while in plants with severe sulfur starvation, lower nitrogen uptake occurs, and non-protein nitrogen compounds are the dominant fractions, e.g., [28,29,30]. Furthermore, the rational application of sulfur fertilizers appears to be one of the important factors responsible for maintaining soil fertility and especially carbon sequestration in soil [19,20]. The widespread observed increases in the amounts of dissolved organic carbon in the surface waters of postglacial landscapes in North America and northern and central Europe have increased in proportion to the rates at which atmospherically deposited anthropogenic sulfur has declined, as indicated by Monteith et al. [31]. The degradation of SOM due to sulfur deficiency in soils is probably responsible for this observation [18]. SOM degradation is an important factor that severely degrades soil quality [32,33] and increases the emission of greenhouse gasses from agricultural soils [34]. The SOM comprises several groups of organic fractions that vary in mass, mobility in the soil profile, and susceptibility to biotransformation. Among these fractions, humic substances are the largest; in particular, humic acids are the most stable carbon pools in the soil and are the most resistant to biodegradation material [35,36,37]. Furthermore, the durability of soil aggregates and the structure of soil are influenced by glomalin-related soil proteins, which are very stable glycol proteins produced in abundance by symbiotic arbuscular mycorrhizal fungi of the genus *Glomus* [38,39]. Another important part of SOM is also the pool of free phenolic acids (FPAs), products of lignin hydrolysis [40] or humic acids, as well as substrates for the formation of humic substances [37,41,42], which, at certain concentrations, can noticeably weaken or stimulate plant growth [43,44]. In addition to the aforementioned fractions, water-soluble organic carbon (WSOC) and hot-water-soluble organic carbon (HWSOC) are important indicators of early changes in both microbial activity and SOM transformations in soils affected by biotic or abiotic factors [45,46,47,48]. Most reports have focused on understanding the effect of sulfur, which is deposited in soils as acid rain, on the dynamics of organic matter in soils and microbial biomass [49,50]. The negative effects of long-lasting NPK-alone mineral fertilization on the quantity and quality of SOM compared to organic manure or organic manure combined with NPK fertilization are well-described phenomena, e.g., [33,51,52,53]. Nitrogen-containing mineral fertilization alone induced soil acidification, an increase in the intensity of recalcitrant C-degrading genes, an increase in the C content of the light fraction of SOM, and a decline in fertility, e.g., [53,54,55,56]. However, available information on the effect of NPKS fertilization on SOM changes compared to NPK alone is limited. A Canadian study, for example, shows that the long-term application of NPK in combination with S resulted in the increased accumulation of soil organic carbon concentrations over NPK alone [20,56,57]. However, the linear decrease in soil sulfate (S-SO_4_) levels of 63% between 2002 and 2014 in topsoil samples collected from fields in Ohio counties in the northwest (NW) did not result in changes in SOM content [7]. It should be noted that comparative studies of the impact of elemental sulfur or various forms of sulfate together with NPKMg fertilization compared to NPKMg fertilization alone on both the SOM composition and the number of microorganisms in soils have not yet been conducted.

Our study provides an opportunity to quantify the effect of the combined application of NPKMg with different forms of S on plant growth and soil. In this study, we determined the influence of different sulfur fertilizers on the yield of perennial ryegrass and the uptake of nitrogen and sulfur. Furthermore, we assumed that the sulfur fertilizers tested would have markedly different effects on the occurrence of bacteria and fungi, the quality and quantity of SOM, and the chemical properties of the soil. Understanding this impact will contribute to the implementation of appropriate agronomic practices to support the protection of SOMs in arable soils. 

## 2. Results and Discussion

### 2.1. The Yield and Uptake of Nitrogen and Sulfur by Perennial Ryegrass

The results of our pot experiment indicate that the addition of sulfur fertilizers to light soils that are poor in this macroelement improved the effectiveness of NPKMg fertilization and therefore resulted in a significant increase in plant yields. The combined application of NPKMg and sulfur fertilizers significantly increased the summary yield of three harvests of ryegrass; the fresh biomass increased in the range of 32.3% to 82.7% (Figure 1; Figure 1A) and the dry biomass increased in the range of 43.7% to 83.3% (Figure 1B) compared to fertilization with NPKMg applied alone. These results are consistent with several reports showing that the appropriate application of sulfur fertilizers is an important factor in improving crop productivity, for example, [11,22,58]. The increase in ryegrass biomass productivity was noticeably related to a significant increase in the uptake of N from 58.3% to 95.5% (Figure 1C) and S from 107.0% to 316.0% (Figure 1D) compared to the control, that is, fertilization with NPKMg applied alone.

The N uptake by ryegrass was not differentially affected by the sulfur forms tested (Figure 1C). However, it was noticed that sulfur uptake was different between the tested forms (Figure 1D). Significantly higher amounts of sulfur uptake by ryegrass were observed in soils supplemented with K_2_SO_4_ and (NH_4_)_2_SO_4_ compared to their uptake from the soil with MgSO_4_ or elemental S applied in the form of Wigor S fertilizer (Figure 1D). This noticeably better growth of ryegrass corresponds especially to the improvement in the nutritional efficacy of nitrogen, which is a well-known phenomenon reported, among others, by Kulczycki [23], Tabak et al. [59], and Zheng and Leustek [24]. Both sulfur and nitrogen are essential elements of amino acids, the basic building blocks of proteins. Furthermore, these macronutrients are important for proper photosynthetic functions because they affect the metabolism and activity of Rubisco, the key catalytic enzyme for carbohydrate production. Therefore, the sufficiency of nitrogen and sulfur in the soil significantly increases the photosynthesis of crops and ultimately leads to an increase in their yield, e.g., [23,24,58]. 

### 2.2. The Levels of Soil Acidity, Nitrogen, Sulfur, and Sulfate

The sulfur fertilizers tested added to the mineral fertilizer (NPKMg) significantly changed some of the chemical properties of the soil (Table 1). The exception is soil acidity, which did not change significantly depending on applied fertilization; soil pH values did not differ significantly. Presumably, lime applied just before ryegrass sowing neutralized the acidification of the soil by sulfur fertilizers, reduced the level of mobile sulfur in the soil, and increased the amounts of sulfur taken up by plants, as reported by several authors, for example, [9,60,61,62]. The total contents of nitrogen, sulfur, and sulfates in soils varied noticeably depending on the type of fertilization (Table 1). 

After 60 days of ryegrass cultivation, the total nitrogen level in the control soil was significantly higher than that in soils with elemental S applied in the form of Wigor S fertilizer, potassium sulfate, or ammonium sulfate. This fact can be explained by the increased accumulation of this element in the above- and below-ground parts of plants grown in soils enriched with various forms of sulfur and, consequently, the improved production of above-ground biomass, which is demonstrated in this work and others, for example, [23,62,63]. Furthermore, an additional application of sulfur fertilizers significantly increased the total sulfur content in the range from 53.2% to 77.6% and sulfates in the range from 167.5% to 475.0% compared to their levels in soil fertilized with NPKMg alone (Table 2). The increase in total sulfur and sulfate content in soils fertilized with NPKMg supplemented with sulfur fertilizers is a well-known phenomenon and has been confirmed by other studies, e.g., [23,62,64,65,66,67]. 

### 2.3. Soil Organic Carbon and Microorganisms

After 60 days of ryegrass cultivation, total organic carbon and the levels of various organic fractions, such as humic acids, free phenolic acids, and GRSPs, as well as WSOC and HWSOC, were analyzed in the soil. The application of elemental sulfur applied in the form of Wigor S fertilizer and sulfates, together with NPKMg fertilizers, modified the contents of most of the aforementioned soil organic carbon fractions, recognized as fertility indicators (Table 2). NPKMg fertilization tested alone and with additional sulfates significantly increased soil carbon content in the range of 18.8% to 38.9% in terms of its level (6.18 g kg^−1^) in the soil before ryegrass cultivation. These can be related to the above-mentioned improved production of underground biomass, which results in higher contents of organic compounds released by the root system and their accumulation. Only the carbon content (5.39 g kg^−1^) in the soil fertilized with NPKMg with elemental S applied in the form of Wigor S fertilizer remained at a level similar to its concentration in the soil before the cultivation of ryegrass and was significantly lower among the objectives tested (Table 2). Furthermore, Gupta et al. [68] reported that the application of elemental S inhibits the accumulation of SOM in the soil. It is difficult to identify the factors and mechanisms of the lower accumulation of total carbon in the presence of elemental S applied in the form of Wigor S fertilizer. This phenomenon is probably the result of the utilization of low-molecular-weight organic compounds, such as root exudates, by organotrophic microbes capable of oxidizing S^0^ [69,70]. Previous studies presented by Stroo and Alexander [71] also suggest that elemental sulfur leads to changes in organic matter solubility, and such changes could influence both the amount of low-molecular-weight organics and organotrophic microbes.

Various impacts of the tested sulfur fertilizers were observed on the contents of humic acids, free phenolic acids, and GRSPs, as well as WSOC and HWSOC (Table 2). It was noticed that none of the sulfur fertilizers applied together with NPKMg significantly changed the humic acid content in the soil samples, although the total carbon content in these soils was lower compared to the control soil fertilized with NPKMg alone. However, the addition of elemental sulfur or sulfate fertilizers to NPKMg significantly decreased the amount of free phenolic acids (FPAs) in the soil compared to the control soil. These findings suggest that the observed decrease in FPAs in these soils may correspond to a weaker degradation of humic substances. Several low-molecular-weight acids, particularly *p*-hydroxybenzoic, vanillic, *p*-coumaric, and ferulic acids, are widespread in soils and, at certain concentrations, can negatively influence plant growth [43]. This decrease in the amount of FPAs in the soils tested could be an additional factor that stimulated the development of ryegrass in our experiment [19,20]. Changes in the content of glomalin-related soil proteins after the application of sulfur fertilizers together with NPKMg are not related to modifications in the levels of total carbon and humic acids. The combined fertilization of the mineral NPKMg with sulfur, potassium, and magnesium sulfates did not affect the content of total glomalin-related proteins (T-GRSPs), as shown in Table 2. Only the application of ammonium sulfate in conjunction with NPKMg fertilization markedly increased the level of T-GRSPs compared to all other soil samples tested (Table 2). The results obtained in this study show that this supplementation with NPKMg should not affect the structure of the soil, and the addition of sulfates should even stabilize soil aggregates. GRSPs are also recognized as indicators directly related to the abundance of arbuscular mycorrhizal fungi [63,72,73]. However, recent reports indicate that the easily extractable glomalin fraction (EE-GRSPs) should be used as a more appropriate biochemical marker of AMF activity [74,75]. The results of this study show that in soils fertilized with sulfates, the EE-GRSP content was significantly higher than in the control soil and soil fertilized with elemental S (Table 2). These findings suggest that sulfate fertilization, even over such a short period, tends to increase the efficiency of mycorrhizae. The application of elemental sulfur is also known to increase the efficiency of mycorrhizae under field conditions, but for longer periods [76]. This observation is in line with our enumeration of the total number of copies of the marker region of fungi in the soils tested. We found that the number of copies of the ITS1 DNA region increased 3.4 and 13.6 times in soil fertilized with potassium sulfate and ammonium sulfate, respectively, compared to soil with NPKMg fertilizer applied alone (Table 3).

The observed stimulative effect of sulfates could be explained by the fact that the uptake of sulfur by AMF is based on the utilization of oxidized forms, such as sulfates [77], and is controlled by the high expression of the sulfite reductase gene [78]. This increase in the total number of copies of the DNA marker region of fungi in soils with sulfates supports the observation mentioned above, indicating an increase in the efficiency of mycorrhizal measures by the content of EE-GRSPs. The observed increase in GRSPs in soils fertilized with additional doses of sulfates should be considered an additional advantage from an agronomic point of view because GRSPs play an important role in the formation and stabilization of micro- and macroaggregates of soil, e.g., [79,80,81]. Furthermore, such stabilized soil aggregates can accumulate water from rainfall, reduce water loss under drought stress, help store organics, and improve the structure of the soil, for example, [82,83]. 

The results of the determination of cold-water-soluble organic carbon (WSOC) and hot-water-soluble organic carbon (HWSOC) also indicated that supplementation of NPKMg with sulfur fertilizers affects the final content of the labile fractions of SOM (Table 3). The WSOC fraction is considered the most mobile and reactive soil carbon source and is the main energy source for soil microorganisms [46,47,84]. The HWSOC fraction contains not only labile C but also other labile nutrients, including the easily mineralizable pool of organic N and extracellular microbial polysaccharides that are recognized as one of the key labile components of organic matter responsible for soil microaggregation [52,64,85,86,87]. The application of magnesium and ammonium sulfates significantly decreased the contents of WSOC and HWSOC after 60 days of ryegrass cultivation compared to the control soil. The HWSOC content was also lower in the soils after the application of elemental S or potassium sulfate than in the control soil (Table 3). The contents of WSOC and HWSOC were significantly correlated (r_p_ = 0.650, *p* < 0.001). Furthermore, the level of HWSOC represented approximately 76.5 % to 78.5% of the total sum of the two estimated organic carbon fractions, as reported by other authors [46,52]. We also observed changes in the content of T-GRSPs, which were significantly negatively correlated with the aforementioned changes in the contents of WSOC (r_p_ = −0.673, *p* < 0.001) and HWSOC (r_p_ = −0.829, *p* < 0.001). However, none of the sulfur fertilizers used in our experiment significantly changed either the total number of copies of the *16S rRNA* gene, which indicates the number of soil bacteria, or the copies of the *nif*H gene, which indicates the number of nitrogen-fixing bacteria, compared to the control sample, that is, NPKMg applied alone (Table 3). Furthermore, it was observed that among the sulfate fertilizers tested, the applied ammonium sulfate significantly increased the number of copies of the *nif*H gene in soil compared to soils fertilized with potassium sulfate or magnesium sulfate only (Table 3). This observation is difficult to explain because sulfur is an essential component of nitrogenase, and this enzyme’s biosynthesis is based on high levels of imported sulfate [88,89]. The aforementioned changes in the content of water-soluble organic carbon fractions, which were not related to changes in the number of molecular markers of bacteria, suggest that there were no significant changes in the number of bacteria, but the microbial utilization of labile carbon nutrients in soil fertilized especially with sulfates was more intensive. An additional reason for the decrease in the contents of WSOC and HWSOC after the application of sulfur fertilizers is the possible accumulation of soil organic carbon compounds in water-stable aggregate fractions stabilized by GRSPs, as described by several authors [73,81,90,91], which can make surfaces hydrophobic [92].

## 3. Materials and Methods

### 3.1. Pot Experiment

Pots were filled with soil collected from the organic horizon of a field in Jelcz Laskowice (51°02′ N 17°20′ E), which was a sandy Arenosol. The texture of the soil was very light, and the granulometric composition of the soil was as follows: sand 86% with dominant medium and fine fractions, silt 12%, and clay 2%. The soil used was very acidic (pH_KCl_ 3.9), and before the setup of the experiment, the acidity of the soil was adjusted to pH_KCl_ 6.62 by lime application at a dose of 2.12 g CaCO_3_ kg^−1^. Total carbon content (C_tot_. 6.18 g kg^−1^), total sulfur content (S_tot_. 115 mg kg^−1^), and sulfates (S-SO_4_ 6.82 mg kg^−1^) were very low. The plant-available phosphorus, potassium, and magnesium contents were as follows: P 74 mg kg^−1^, K 78 mg kg^−1^, Mg 13 mg kg^−1^, and S-SO_4_ 6.82 mg kg^−1^. Each plastic pot (volume 2.5 L, height 14.5 cm, circumference 19 cm) was filled with 2.5 kg of tested soil, previously air-dried, sieved (<2 mm), and deacidified, as mentioned above, two weeks before the setup of the experiment. Macro- and microelements were added before sowing by mixing them with the entire soil mass of the pot. The amounts of applied doses of macronutrients (N, P, K, Mg) were balanced, taking into account the amount added with tested sulfates so that they were even and the same in all pots. Macronutrients were applied to the soil in the following doses: N 104 mg kg^−1^ as ammonium nitrate, P 124 mg kg^−1^ as calcium dihydrogen phosphate, K 293 mg kg^−1^ as potassium chloride, and Mg 92 mg kg^−1^ as magnesium oxide. The experiment was set up with the same doses of sulfur at a level of 120 mg S kg^−1^ of soil compared to the soil without sulfur fertilization. The following groups were established: NPKMg alone—control soil without sulfur fertilization; NPKMg + S^0^—soil fertilized with additional elemental sulfur in the form of Wigor S fertilizer, which is granulated 90% elemental sulfur with 10% bentonite (“Siarkopol” Tarnobrzeg Ltd., Poland); NPKMg + K_2_SO_4_—soil fertilized with additional potassium sulfate; NPKMg + MgSO_4_—soil fertilized with additional magnesium sulfate; NPKMg + (NH_4_)_2_SO_4_—soil fertilized with additional ammonium sulfate. Calcium carbonate, macronutrients, and sulfate salts used were laboratory-grade chemicals purchased from the Chempur Company (Piekary Śląskie, Poland). The perennial ryegrass (*Lolium perenne* L.) cultivar Solen was used in the pot experiment. After mineral fertilization, 0.3 g of perennial ryegrass (~100 seeds) per pot was seeded in 20 holes and covered with a ~0.5 cm layer of soil. Grass cuts were made at 30, 45, and 60 days of vegetation. Each group was set up with four replications (pots). The experiment was carried out in a completely randomized design after each watering. During the vegetation period, constant soil moisture based on weight loss was kept at 50% of the water-holding capacity by adding deionized water every two days. This level of moisture ensures the proper proportions of the air–water phases. After each cut, soil moisture was adjusted to 50% of the water retention capacity. The pots with plants were kept in a controlled growth chamber with a photoperiod of 16 h/8 h light/dark and 26–28 °C/16–18 °C day/night temperatures. 

### 3.2. Sample Preparation

The green biomass yields of the ryegrass were determined immediately after cutting, and the dry biomass yields were determined after drying at 105 °C to a constant weight. After the third cutting of the above-ground parts of the plants, the bulk soil samples were separated into 3 fractions. One part was immediately frozen and then lyophilized for further analysis. The lyophilized soil samples were sieved through a 2 mm sieve and stored at 72 °C. The second part was immediately used for organic carbon fraction analyses, and the remaining bulk soil samples were dried (+110 °C), sieved (<2.0 mm), and stored at room temperature for physicochemical analyses.

### 3.3. Nitrogen and Sulfur Analyses

The nitrogen content in the dried biomass was determined using the Kjeldahl method [93], which involves wet digestion and distillation, and the total sulfur content was determined with the Butters and Chenery method [94]. Soil acidity was determined in 1:2.5 soil/1 M KCl suspensions using a CP505 digital pH-meter (Elemetron Co., Zabrze, Poland). The total contents of C, N, and S in the soil samples tested were determined by the combustion method using a TruSpec analyzer (Leco Co., Benton Harbor, MI, USA). In turn, the amount of S-SO_4_ in soil samples tested was estimated using the Bardsley and Lancaster method [95].

### 3.4. Soil Organic Carbon Analyses

#### 3.4.1. Humic and Free Phenolic Acids 

Humic acids (HAs) were extracted from 30 g of lyophilized soil samples using the modified Swift method [96], as previously described [23]. The determination and extraction procedure of the free phenolic acid content in the tested soils was carried out according to the method described by Krygier et al. [97] with some modifications. Lyophilized soil samples (10 g) were extracted three times with 80% methanol. The extract solvent was successively removed by vacuum evaporation on a rotavapor. The residue obtained was dissolved in methanol. Subsequently, 0.5 mL of FC reagent, 1 mL of a 20% (*w*/*v*) solution of anhydrous sodium carbonate, 0.2 mL of phenolic acid methanol extract, and 8.3 mL of water were combined. The mixture after incubation (60 min at room temperature in darkness) was centrifuged (10,000× g, 10 min, 4 °C). The absorbance of the samples at 725 nm was read against the blank reagent. The results were expressed as micrograms of sinapic acid equivalents per gram of lyophilized soil on the sinapic acid calibration curve. 

#### 3.4.2. Glomalin-Related Soil Proteins

Total glomalin-related soil proteins (T-GRSPs) and easily extractable glomalin-related soil proteins (EE-GRSPs) were isolated from lyophilized soil samples in 50 mM citrate buffer (pH 8.0) and 20 mM citrate buffer (pH 7.0), respectively [98]. Soil samples (10 g) were covered with appropriate buffers and autoclaved at 121 °C for 60 min to extract T-GRSPs and at 121 °C for 30 min to extract EE-GRSPs. The extraction was carried out several times until the total organic fraction of the soil was washed out. After each autoclaving, the buffer containing GRSPs was removed, and the soil samples were again covered with a sterile buffer. The extracts collected after each heating were combined and supplemented to equal volume for each sample and centrifuged at 10,000 rpm for 10 min. The GRSP content in the supernatants was quantified using the Bradford method using bovine serum albumin (Sigma-Aldrich, Inc., Saint Louis, MO, USA) as a standard.

#### 3.4.3. Water-Soluble Organic Carbon and Hot-Water-Soluble Organic Carbon

The amounts of water-soluble organic carbon (WSOC) and hot-water-soluble organic carbon (HWSOC) were determined in fresh soil samples using the Haynes and Francis method [99] modified by Ghani et al. [52], as previously described [23].

### 3.5. Microbial DNA qPCR Assays

The nonspecific quantitative PCR (qPCR) method was used to determine the total number of fungi and bacteria, as well as to quantify the nitrogen-fixing bacterial gene (*nif*H). Total soil DNA was extracted using the Syngen Soil DNA Mini Kit according to the manufacturer’s instructions (Syngen Biotech, Wrocław, Poland). DNA concentration and purity were measured spectrophotometrically at wavelengths of 230, 260, 280, and 320 nm with the Eppendorf BioPhotometer (Eppendorf AG, Hamburg, Germany). To estimate the total amount of bacterial DNA, a *16S rRNA* gene fragment of 136 bp was amplified with the universal primers 926F (5′-AAA CTC AAK GAA TTG ACG G) and 1062R (5′- CTC ACR RCA CGA GCT GAC) under the following reaction conditions: initial denaturation and polymerase activation for 12 min at 95 °C, denaturation for 20 s at 95 °C, annealing for 30 s at 62 °C, and elongation for 20 s at 72 °C. Furthermore, PCR amplification of a 250 bp long section of the fungal ITS1 DNA region with the primers ITSFI2 (5′-GAA CCW GCG GAR GGA TCA) and 5.8S (5′-CGC TGC GTT CTT CAT CG) was performed using the following conditions: initial denaturation for 12 min at 95 °C, denaturation for 20 s at 95 °C, annealing for 30 s at 66 °C, and elongation for 30 s at 72 °C. The number of nitrogen-fixing bacteria was also evaluated with primers specific for the 457 bp long fragment of the *nif*H gene: *nif*H-F (5′-AAA GGY GGW ATC GGY AAR TCC AC) and *nif*H-R (5′-TTG TTS GCS GCR TAC ATS GCC ATC AT); this was carried out under the following conditions: initial denaturation for 12 min at 95 °C, denaturation for 20 s at 95 °C, annealing for 30 s at 64 °C, and elongation for 50 s at 72 °C. Reactions were carried out for 40 cycles in final volumes of 20 µL, using 1 µL of DNA template (at concentrations of 0.1 pg to 10 ng µL^−1^), 0.5 L of each primer (at a concentration of 10 pmol µL^−1^), and 4 µL of the 5x HOT FIREPol EvaGreen qPCR mix (Solis BioDyne, Tartu, Estonia). DNA copy numbers were calculated with Q-qPCR software from Q-qPCR QuantaBio Thermal Cycler (QuantaBio, Beverly, MA, USA) based on standard curves prepared with the following reference strains: *Pseudomonas* sp. strain X (for analysis of the bacterial *16S rRNA* gene), *Fusarium* sp. 5 (for fungal ITS1 DNA analysis), and *Sinorhizobium meliloti* 1021 (for *nif*H gene analysis). Referential strains were also used to establish the conditions of the PCR reaction using the classical gradient PCR method with the Mastercycler Gradient Thermal Cycler (Eppendorf AG). To prepare the standard curves, DNA from the pure culture of the reference fungal strain was extracted using the Syngen Fungi DNA Mini Kit (Syngen Biotech), while for the bacterial strains, the GeneElute Bacterial genomic DNA Kit (Sigma-Aldrich) was used according to the manufacturer’s instructions. The DNA obtained was used as a template for classical PCR analyses. The reaction conditions were the same as those of qPCR, except for the additional final elongation step performed for 10 min. The PCR products were visualized by electrophoresis on an agarose gel with ethidium bromide. The appropriate PCR product bands were excised/cut out using a UV transilluminator, and their DNA was purified with the Syngen Gel/PCR Mini Kit (Syngen Biotech). DNA concentrations were determined spectrophotometrically, and the copy numbers were calculated with the online DNA Copy Number and Dilution Calculator from Thermo Fisher Scientific Corporation. Finally, the DNA reference samples were serially diluted and used as qPCR standards.

### 3.6. Statistical Analysis

All tests were carried out in four replications for each sample. The yield sizes and results of the chemical analysis were subjected to a one-way variance analysis. Before performing the variance analysis, the homogeneity test of variance within groups was performed using the Levene test and the Shapiro–Wilk test of the correspondence of variables with the normal distribution. The relevance of the mean differences was evaluated using the Tukey post hoc test with a significance level of *p* ˂ 0.05. The dependencies between characteristics were determined by calculating Pearson’s correlation coefficient. The statistical program R [100] was used for all statistical analyses. 

## 4. Conclusions

The pot experiment with perennial ryegrass in sulfur-deficient soil provided data to evaluate the effect of different sulfur fertilizers applied together with NPKMg fertilization on soil organic matter. The results of this study indicate that the application of additional sulfur with NPKMg significantly stimulated the growth of ryegrass as a result of the increase in the uptake of nitrogen and sulfur by plants, which is a well-known phenomenon described in the literature. The application of sulfur, especially in the form of sulfates, changes the composition of the SOM. The decrease in the contents of FPAs and water-soluble organic compounds and the increase in the content of GRSPs are probably a result of the better efficiency of the mycorrhizae. These facts indicate that sulfur fertilizers stabilize SOM and confirm the hypothesis that these fertilizers support the maintenance of soil fertility. This effect is noticeable even a short time after soil fertilization based on analyses of labile soil organic carbon fractions.

Taking into account all of the observed effects and the acknowledged positive impact of sulfur and sulfate additives on the efficacy of NPKMg fertilizers and soil properties, the results of this study represent a promising agronomic option to increase the environmental sustainability of agricultural systems. Further studies considering appropriate dosages of sulfur or sulfates or their mixtures with NPKMg fertilization require a rigorous investigation of the properties of SOM fractions and the biological activity of soils under field conditions with different crops. 

## Data Availability

The data presented in this manuscript are available from the authors upon reasonable request.

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
