# Peer review of "The Effect of Various Forms of Sulfur on Soil Organic Matter Fractions and Microorganisms in a Pot Experiment with Perennial Ryegrass (Lolium perenne L.)"

_plants, 2023, doi:10.3390/plants12142649_

Round 1

Reviewer 1 Report

The author in this paper try to quantify the effect of the combined application of NPKMg with different forms of S on plant growth and soil. The study is vital to the implementation of appropriate agronomic practices to support the protection of SOMs in arable soils. The work authors have done is of interest and the language of the manuscript is acceptable for the readers. The structure of the paper is logical as well. Still, I think the authors should consider some other questions below. Overall, the manuscript can be accepted after moderate revision. The main issues are listed as follows:

1. In line 80-81. I think the authors need to state more about the effect of other NPK fertilization on SOM and soil microbial processes.  

2. In line 276-277. According to this analysis on soil pH value, how should the application of sulfur fertilizers be adjusted when the soil acidity is too high?

3. Based on the conclusions, I think there's another limitation of this paper. Since the formation of the soil microbial environment is a long-term process. Maybe for several growths period, can the stable soil microbial established? And so as to the format of soil structure. So the long-term field trial should be considered.

Author Response

Dear Reviewer,

We appreciate your comments and suggestions which noticeably help support the improvement of our manuscript. I have provided my explanations dealing with your comments below.

1. In lines 80-81. I think the authors need to state more about the effect of other NPK fertilization on SOM and soil microbial processes.  

Adequate pieces of information were added based on available research data (lines 98 - 103), (all corrections made are highlighted in color);

2. In lines 276-277. According to this analysis of soil pH value, how should the application of sulfur fertilizers be adjusted when the soil acidity is too high?

For the purpose of this study, we limed the soil with a reasonably high amount of calcium carbonate two weeks before the set-up of the experiment which created an evident buffer capacity of the soil as described in the section of Materials and Methods (current lines 133-135). Such treatment is good practice in agriculture in Central Europe. 

3. Based on the conclusions, I think there's another limitation of this paper. Since the formation of the soil microbial environment is a long-term process. Maybe for several growths period, can the stable soil microbial established? And so as to the format of soil structure. So the long-term field trial should be considered.

We agree that long-term field trials based on several growth periods will be more informative and reliable. But we try to evaluate whether is it possible to analyze the effects of sulfur fertilizers using the study of labile carbon organic fractions, which are sensitive to various factors in a short period. Moreover, long-term field experiments are planned to estimate the action of different mineral forms of sulfur on the soil microbiome and soil organic fractions.  

Reviewer 2 Report

The manuscript entitled "The Effect of Various Forms of Sulfur on Soil Organic Matter Fractions and Microorganisms in a Pot Experiment with Perennial Ryegrass (Lolium perenne L.)" is submitted to Plants MDPI. The study is interesting, and the manuscript is written well. There are some points that should be considered, and I hope they can be helpful for authors to improve their work.

Line 62: SOM degradation of SOM

Lines 75–79: Please concise the sentence.

The introduction is informative, although it can be improved by adding more mechanistic statements about the effects of sulfur on soil and crop productivity.

Please mention the source of the NPKMg applied.

Please introduce elemental sulfur in the form of Wigor S fertilizer appropriately.

Materials and methods are well detailed.

There is no information about the uptake of sulfur by perennial ryegrass. Additionally, discussion of N is not enough.

Results are presented well; however, discussion can be improved.

The biggest concern is that talking about organic matter in the soil after a 60-day experiment might not be so reasonable. As a fact, organic matter is stable, and it’s hard to see the effect of different sources of S on soil organic matter.

The conclusion is well written.

Manuscript is well-written and minor editing of English language required. 

Author Response

Dear Reviewer, 

Thank you very much for your valuable comments and suggestions that helped us to rewrite my manuscript. I have placed my explanations dealing with your comments below; all changes made to the manuscript are highlighted in color;

  1. Line 62: SOM degradation of SOM was replaced with “The degradation of SOM” (current line 80).
  2. Lines 75–79: Please concise the sentence.

This sentence was rewritten and shortened to be more concise  (current lines 93-96). 

  1. The introduction is informative, although it can be improved by adding more mechanistic statements about the effects of sulfur on soil and crop productivity.

In the revised version of our manuscript, the paragraphs dedicated to transformations of sulfur and its impact on soil and crop productivity were extended; (current lines 52-56; 59-61; 66-74). 

  1. Please mention the source of the NPKMg applied.

This information, as well as the names of the chemical compounds in which these macroelements (NPKMg) were used, have been supplemented.; (currently lines 138-141 and 148-150). The supplier of used chemicals is also added. 

  1. Please introduce elemental sulfur in the form of Wigor S fertilizer appropriately.

This suggestion has been followed throughout the manuscript.

6. There is no information about the uptake of sulfur by perennial ryegrass. Additionally, a discussion of N is not enough.

Information about the uptake of sulfur by the tested plant was described by us in the original version of the manuscript (lines 265-269), the current version (lines 293-296); unfortunately, the wrong figures (1C and 1B instead of 1D) were assigned to this description; these mistakes were corrected.

In the revised version of the manuscript information about the nitrogen cycle, its transformations, and nutritional efficacy was added to the section of Results and Discussion (current lines 299-305; 441-446);

7. Results are presented well; however, discussion can be improved.

The section of Results and Discussion was supplied with an explanation of the improvement of the nutritional efficacy of nitrogen and additional references were added (current lines 299-305; 441-446) and all errors were corrected. 

8. The biggest concern is that talking about organic matter in the soil after a 60-day experiment might not be so reasonable. As a fact, organic matter is stable, and it’s hard to see the effect of different sources of S on soil organic matter.

We agree that a long-term field experiment and several growth periods are more informative and reliable than a pot experiment, especially for a study of microbial metabiome and turnover of SOM under conditions. We try in our study to evaluate the possibilities of analysis of sulfur fertilizer effects on SOM based on the changes of the labile soil carbon fractions. According to several reports, the study of organic labile fractions is an important indicator of early changes in both microbial activity and SOM transformations in soils affected by biotic or abiotic factors even stable carbon fractions have not been changed. 

Further field studies are planned to verify our pot experiment's results and estimate the impact of different inorganic forms of sulfur on soil organic matter and bioactivity.

Round 2

Reviewer 1 Report

  • I recommend accepting the revised manuscript.

Reviewer 2 Report

The manuscript has been improved by authors.